# Integrating Commercial-Off-The-Shelf Components into Radiation-Hardened Drone Designs for Nuclear-Contaminated Search and Rescue Missions

Arjun Earthperson [ID] and Mihai A. Diaconeasa *[ID]

Department of Nuclear Engineering, North Carolina State University, Raleigh, NC 27695, USA
* Correspondence: madiacon@ncsu.edu; Tel.: +1-(919)-515-3768

**Abstract:** This paper conducts a focused probabilistic risk assessment (PRA) on the reliability of commercial off-the-shelf (COTS) drones deployed for surveillance in areas with diverse radiation levels following a nuclear accident. The study employs the event tree/fault tree digraph approach, integrated with the dual-graph error propagation method (DEPM), to model sequences that could lead to loss of mission (LOM) scenarios due to combined hardware–software failures in the drone's navigation system. The impact of radiation is simulated by a comparison of the total ionizing dose (TID) with the acceptable limit for each component. Errors are then propagated within the electronic hardware and software blocks to determine the navigation system's reliability in different radiation zones. If the system is deemed unreliable, a strategy is suggested to identify the minimum radiation-hardening requirement for its subcomponents by reverse-engineering from the desired mission success criteria. The findings of this study can aid in the integration of COTS components into radiation-hardened (RAD-HARD) designs, optimizing the balance between cost, performance, and reliability in drone systems for nuclear-contaminated search and rescue missions.

**Keywords:** dual-graph error propagation model; discrete dynamic event tree; dynamic probabilistic risk assessment; error propagation; OpenEPL; OpenPRA; COTS





## 1. Introduction

Uncrewed Aerial Vehicles (UAVs), commonly known as drones, have become increasingly popular in various applications due to their versatility and cost-effectiveness. One such application is search and rescue (SAR) activities, particularly in the aftermath of a nuclear accident. These missions are critical to assess the extent of damage, locate survivors, and monitor radiation levels [1,2]. However, the high radiation levels in such environments pose a significant risk to the electronic components of UAVs, potentially leading to mission failure. The use of commercial off-the-shelf (COTS) drones in SAR missions offers several advantages, such as cost-effectiveness, rapid deployment, and ease of operation. However, these drones are not typically designed to withstand the harsh radiation environments encountered in nuclear accidents. On the other hand, radiation-hardened UAVs may be employed, but their use may not be justified in all cases due to factors such as availability, cost, and mission-specific requirements.

Given these challenges, there is a pressing need to develop a systematic approach to assess the reliability of COTS drones in nuclear SAR missions. This paper presents a probabilistic risk assessment (PRA) approach to determining radiation-hardening limits for COTS drones in radiological SAR operations based on predefined mission success criteria. PRA offers a comprehensive framework that is well-suited to modeling intricate dependencies and failure modes in complex systems. These methods have significantly contributed to ensuring safety in current nuclear and aerospace operations.

*Literature Review*

Recent years have seen a surge in the use of drones for SAR missions, with studies such as that by Murphy et al. [3] highlighting their potential in providing real-time situational awareness during disaster management. The effects of radiation on electronic components are also well-documented. Fleetwood [4], Normand [5] and Dodd et al. [6] provide comprehensive reviews of these effects, including single-event upsets (SEUs), and total ionizing dose (TID) effects, which are particularly relevant to our study. As such, the use of COTS components in radiation environments presents both challenges and opportunities. Barnaby et al. [7,8] underscore the need for radiation-hardening in microelectronics, a concept explored in depth by Ladbury [9]. On the other hand, PRA has been widely used to evaluate the reliability of systems, as outlined by Apostolakis [10] and Modarres et al. [11]. However, the application of PRA to drone systems, particularly in the context of nuclear SAR missions, is a novel research area. While the literature provides valuable insights into the individual aspects of drone design for nuclear-contaminated SAR missions, there is a need for a comprehensive study that integrates these aspects into a systematic approach for assessing drone system reliability. This study aims to fill this gap by proposing a PRA-based approach for assessing the reliability of COTS drones in nuclear SAR missions.

Our proposed approach utilizes event trees, fault trees, and a Markov-chain-based dual-graph error propagation methodology to model sequences leading to Loss of Mission (LOM) due to component failures in the drone's navigation system. Radiation effects are simulated by calculating the total ionizing dose (TID) against the permissible limit per component, and errors are propagated within the electronic hardware and software blocks to quantify navigation system availability per radiation zone. The results provide a demonstrative assessment of the drone's availability in nuclear SAR missions.

The remainder of this paper is organized as follows: Section 2 provides a brief overview of the problem scope and the methodology behind the proposed solution. This includes the proposed PRA approach, including the modeling of radiation effects on electronic components and the integration of dynamic failure scenarios. Section 3 presents a case study demonstrating the application of the proposed approach to a COTS drone navigation system. Section 4 discusses the results and their implications for drone design, component selection and radiation-hardening. Finally, Section 5 concludes the paper by addressing the limitations and outlining ideas for future research.

## 2. Methodology

In this section, we provide a brief overview of the problem scope and the methodology behind our proposed solution to the assessment of COTS drone availability in nuclear SAR missions. We introduce a few terms within resilience ontology in the context of temporal logic using the Kripke structure notation to model time-dependent risk [12–15]. The formal equation and definitions are introduced in Equation (1) and Table 1.

**Table 1.** Three-tuple Kripke terms for a three-state transition system.

| Term | Definition | Description |
|------|-----------|-------------|
| $S$ | $S = \{nominal,\ degraded,\ failure\}$ | A set of possible states. |
| $i$ | $i \subseteq S = \{nominal\}$ | The initial state, which is nominal. |
| $R$ | $R \subseteq S \times S$ | A mapping or transition relation, where $R$ is left-total (if the source set $X$ equals the domain, $R \subseteq X \times Y$ is left-total), and $M$ is fully connected. |

A system $M$ consists of states $S$ that transition along a bounded but countably infinite long path $P$, starting with the initial state $i$. $R$ is the transition relation that maps all the valid state transitions. An equivalent definition can be expressed using the edge list $e = \{A,\ B,\ C,\ D,\ E,\ F,\ G,\ H,\ I\}$, which forms our alphabet. The traversal of path $P$

produces words $W$. This allows for us to build a grammar with which we can define events or system properties, and emergent behaviors. This grammar can include words of our choosing; some examples are listed in in Table 2. By extension, each trajectory or word is expressible in a temporal sense, as depicted in Figure 1 [16].

$$M := \langle S, i, R \rangle \tag{1}$$

**Table 2.** State transition definitions for the three-state model referenced in Figure 1.

| Regular Expression | Term | Description |
|---|---|---|
| $A^+$ | Ideal/Perfect System | No errors, faults, or failures occur. |
| B | Fault | A fault is a weakness that can potentially lead to errors. |
| $E^+ \mid C^+$ | Error Propagation | A move from an initial error state leads to a subsequent one. |
| D $\mid$ H | Failure | System fails from either a degraded or a nominal state. |
| I $\mid$ G $\mid$ F | Recoverable System | Move from higher to lower degradation. |
| B($C^* \mid$ I) | Fault-Tolerant | Avoid transition to failure, given a fault. |
| $A^+ \mid$ (B($C^* \mid$ I)) | Failure-Avoidant | No failures occur. |
| G $\mid$ F | Resilient System | Recover from a failure, either fully or partially. |
| B($C^* \mid$ D($E^* \mid$ G)) $\mid$ (H($E^* \mid$ G)) | Irrecoverable System | Neither completely fails, nor returns to nominal. |
| B$C^*(\epsilon \mid$ D$E^*)$ | Permanently Failed | System remains irrecoverable forever. |

($\epsilon$): empty set, ($x^*$): zero or more instances of x, ($x^+$): one or more instances of x.

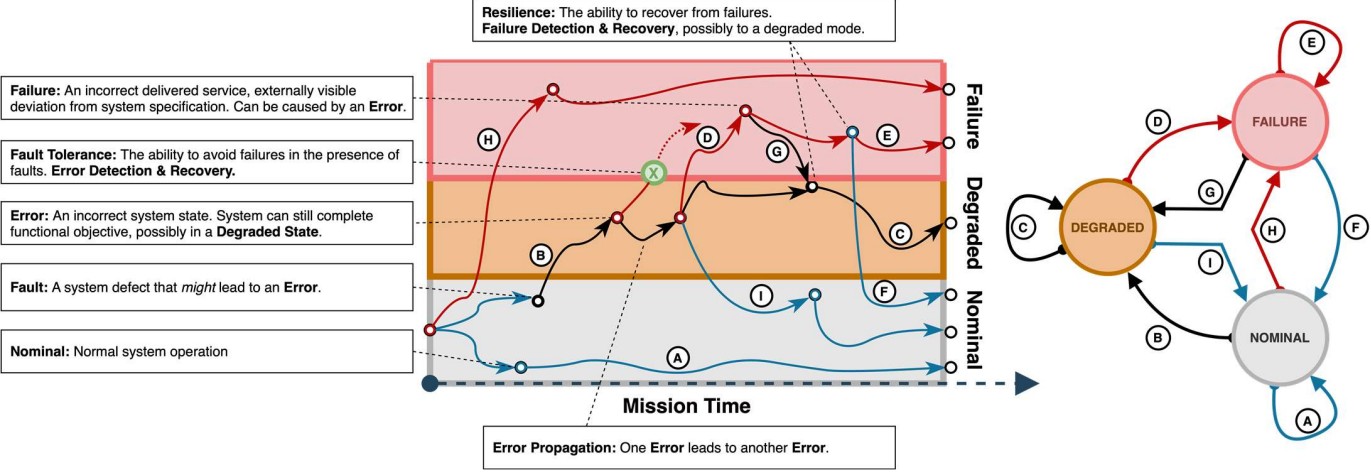

**Figure 1.** State transitions within a three-state system, initialized as nominal. (**left**) temporal, (**right**) state machine.

Risk is evaluated by identifying potential failure scenarios, assessing their likelihood, and determining the consequences if these failures occur. Consequently, risk is formally expressed as a complete set of $N$ triplets that include a scenario description $s_i$, its probability $p_i$, and the consequences, i.e., the resulting damage measure or evaluation metric $x_i$:

$$\mathcal{R} = \{ \langle s_i, p_i, x_i \rangle \}_c, \quad i = \overline{1, N} \tag{2}$$

Conventional PRA approaches involve sequence-based modeling where initiating events are chosen and conditional event progressions are analyzed, leading to end states of interest. By incorporating consequence information into these PRA models, frequency–consequence curves can be formulated [17]. In event tree analysis, probabilities are assigned to functional events depicting various components, systems, or operator actions using

fault trees [18]. These probabilities take into account either time-dependent or on-demand failure modes given predetermined mission durations.

In certain situations, however, event tree/fault tree methods may need to be supplemented with specialized analysis techniques to model the systems that involve error propagation failure modes, or incorporate multiple failure paths, such as the example provided in the previous section [19]. Tracking the propagation of errors from discrete sub-components to system or functional levels in such systems presents a unique challenge. DEPM is an extension to Markov chains, which enables the modeling of data flows and control flows as two separate Markov chains within a cyber-physical system [20]. DEPM terms and definitions are listed in Equation (3) and Table 3, respectively. The interaction between the control and data Markov chains is specified by the DEPM algorithm. In DEPM, we consider a system as independent, and discrete elements, which can be sensor modules or software blocks in a mechatronic system [21]. When a fault activates, it can propagate to connected elements, which may corrupt data, or alter the control and data flows. We present the DEPM formalism along with an example in the following figures.

$$DEPM := \langle E, D, A_{CF}, A_{DF}, C \rangle \tag{3}$$

**Table 3.** Definitions for terms in dual-graph error propagation model (DEPM).

| Term | Definition |
|------|------------|
| $E$ | A set of elements, always non-empty. |
| $D$ | A set of optional data terms. |
| $A_{CF}$ | An edge-list representing control flows. |
| $A_{DF}$ | An edge-list representing data flows. |
| $C$ | A list of conditional expressions, which apply to the element set $E$. |

Flows from an element may branch erroneously, depending on its corresponding failure rates/probabilities. By extension, error propagation analyses can simulate single-event upsets (SEUs). To perform quantitative evaluations, DEPM models are automatically transformed into continuous time (CTMCs) or discrete-time Markov Chains (DTMCs). Figure 2 and Table 4 illustrate an example DEPM with associated conditional logic expressions.

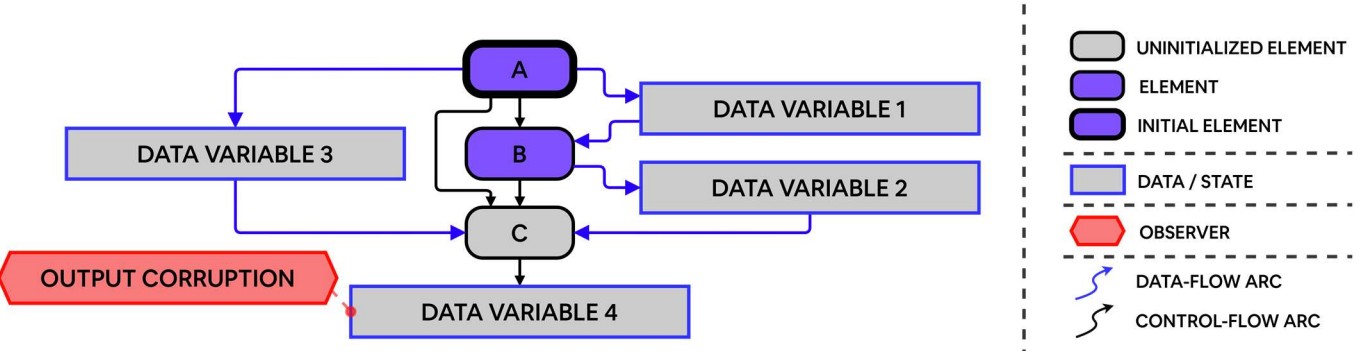

**Figure 2.** Example DEPM with a legend.

The DEPM model in Figure 2 depicts the execution of a serial code. Assembly operations, represented as elements A, B, and C, read and write data variables 1 and 2, and to and from CPU registers. Element A changes variables 1 and 3. Elements B and C change variables 2 and 4. Element B reads from data variable 1, while element C reads from both variable 2 and variable 3.

This DEPM computes the probability that the output variable, data variable 4, is corrupted. Given that SEUs are stochastic in nature, this may occur at any time [22]. To achieve this goal, expressions can be evaluated by employing quantifiable Boolean formulae (QBF) evaluating satisfiability solvers [23,24]. Relevant metrics such as the mean time to failure (MTTF), the number of total failures, and time-dependent failure probability can be directly quantified using formal verification and model-checking methods. Since it is based on probabilistic modeling checking, DEPM is better-suited to modeling the behavior of smaller, but highly interdependent systems than traditional methods such as fault trees and Markov chains.

**Table 4.** Conditional logic table for example DEPM in Figure 2.

| Element | Conditional Expressions |
|---|---|
| A | **always:**<br>    **with** P(0.8): DATA VARIABLE 1, DATA VARIABLE 2 = error<br>    **with** P(0.2): DATA VARIABLE 1, DATA VARIABLE 2 = ok |
| B | **if** DATA VARIABLE 1 = error, **then:**<br>    **with** P(0.9): DATA VARIABLE 2 = ok<br>    **with** P(0.1): DATA VARIABLE 2 = error<br>**else:**<br>    **with** P(1.0): DATA VARIABLE 2 = ok |
| C | **if** DATA VARIABLE 2 & DATA VARIABLE 3 = ok, **then:**<br>    **with** P(1.0): DATA VARIABLE 4 = ok<br>**else:**<br>    **with** P(0.2): DATA VARIABLE 4 = ok<br>    **with** P(0.8): DATA VARIABLE 4 = error |

## 3. Demonstration Case

In this demonstration case, we consider a COTS drone equipped with a navigation system, communication system, and radiation sensor payload. The drone's primary mission is to perform SAR activities in a nuclear-contaminated environment, which includes monitoring radiation levels, identifying damaged infrastructure, and locating survivors. The drone's navigation system comprises a power subsystem, inertial measurement unit (IMU) sensors, positioning sensors, and a Kalman filter. The drone is tasked with flying over a predefined search area, which is divided into three zones with varying radiation levels. The drone starts its mission in the low-radiation zone (Zone A), transitions to the medium-radiation zone (Zone B), and finally enters the high-radiation zone (Zone C) before returning to the base. Radiation levels in Zone A are based on background radiation [25,26]. Radiation levels in Zone B and C are sourced from samples in and around Unit 4 and surrounding buildings at the Chernobyl nuclear power plant (NPP) shortly after explosion [27]. Since mission success is dependent on the UAV successfully performing SAR activities for each zone before Loss of Vehicle (LOV) occurs, the analysis is finished once the drone completes its mission objectives in Zone C. The absorbed dose rates, time in each zone, and total absorbed dose are sampled from a truncated normal distribution ($\mathcal{N}(\mu, \sigma)$—normal distribution, truncated to represent a realistic and physically meaningful sampling space; for example, time cannot be negative) and a loguniform distribution ($\mathcal{LU}(min, max)$—loguniform distribution with *min* and *max*), listed in Table 5 and illustrated in Figure 3. The proposed approach is applied to assess the drone's availability in each radiation zone by considering the potential failure scenarios due to radiation-induced component failures. TID is calculated for each component in the drone's navigation system and compared with the component's permissible limit to determine the likelihood of failure.

**Table 5.** Ambient radiation dose rates for radiation zones A, B, C.

| Zone | Dose Rate [rad/hour] | Elapsed Time [minute] | Total Received Dose [rad] |
|:---:|:---:|:---:|:---:|
| A | $\mathcal{N}(\mu \approx 2.50 \,,\, \sigma \approx 1.50) \times 10^{-5}$ | $\mathcal{N}(\mu \approx 6.44 \, min \,,\, \sigma \approx 3.97 \, min)$ | $\mathcal{N}(\mu \approx 3.41 \,,\, \sigma \approx 2.52) \times 10^{-6}$ |
| B | $\mathcal{LU}(min = 0.30 \,,\, max = 3.00) \times 10^{-2}$ | $\mathcal{N}(\mu \approx 51.4 \, min \,,\, \sigma \approx 6.38 \, min)$ | $\mathcal{N}(\mu \approx 8.79 \,,\, \sigma \approx 5.73) \times 10^{0}$ |
| C | $\mathcal{N}(\mu \approx 2.52 \,,\, \sigma \approx 0.98) \times 10^{4}$ | $\mathcal{N}(\mu \approx 61.7 \, min \,,\, \sigma \approx 7.93 \, min)$ | $\mathcal{N}(\mu \approx 4.33 \,,\, \sigma \approx 2.70) \times 10^{3}$ |

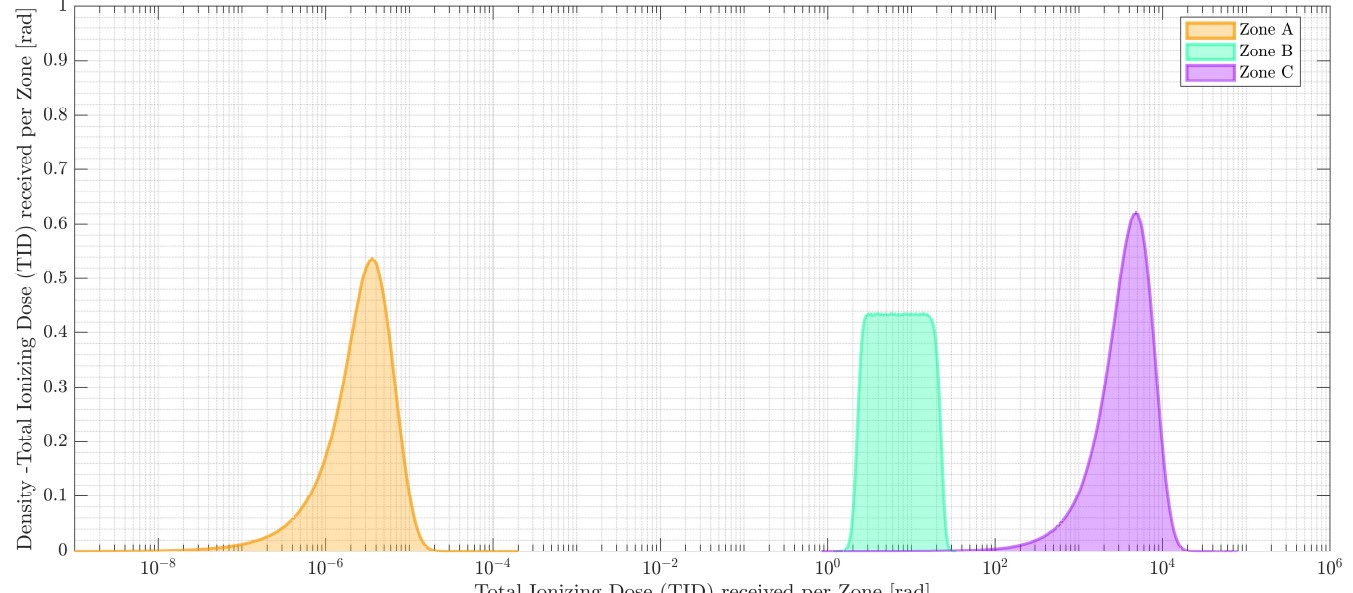

**Figure 3.** Kernel density estimates for total received dose by zone.

### 3.1. Scenario Description

The event tree for the drone's mission is constructed based on the sequence of events that the drone is expected to encounter during its mission. Starting with the initiating event, the likelihood of navigation system availability is computed for zones A, B, and C. At each functional event, the tree branches into two outcomes: success or failure. The success branch leads to the next event in the sequence, while the failure branch leads to a Loss of Mission (LOM) end state. The failure probabilities at each node are calculated using the navigation system fault tree. Figure 4 illustrates this event tree, modeled in the OpenPRA framework [28].

### 3.2. Assumptions and Simplifications

In order to focus our analysis on the presented methodology, we made several assumptions and simplifications. These are necessary to streamline the discussion and concentrate on the core concepts, but it is important to note that they may limit the comprehensiveness of the model.

The event tree depicted in Figure 4 only considers the availability of the navigation system. A more comprehensive model would consider all components of the UAV and their interdependencies, including the potential for common cause failures (CCFs). As a result, the baseline failure probabilities presented in this study may appear lower than they would in a more complex model that includes CCFs. Additionally, when mechatronic systems are exposed to radiation environments, they can fail due to a variety of mechanisms. Expressing failure likelihoods in terms of TID effects abstracts away from the underlying failure mechanisms and simplifies the associated failure modes. Ionizing radiation can accelerate degradations in digital hardware through single-event upsets, gate oxide breakdown, and hot carrier injection, amongst others. In this study, we only consider TID effects, which are the cumulative effects of ionizing radiation on materials and devices.

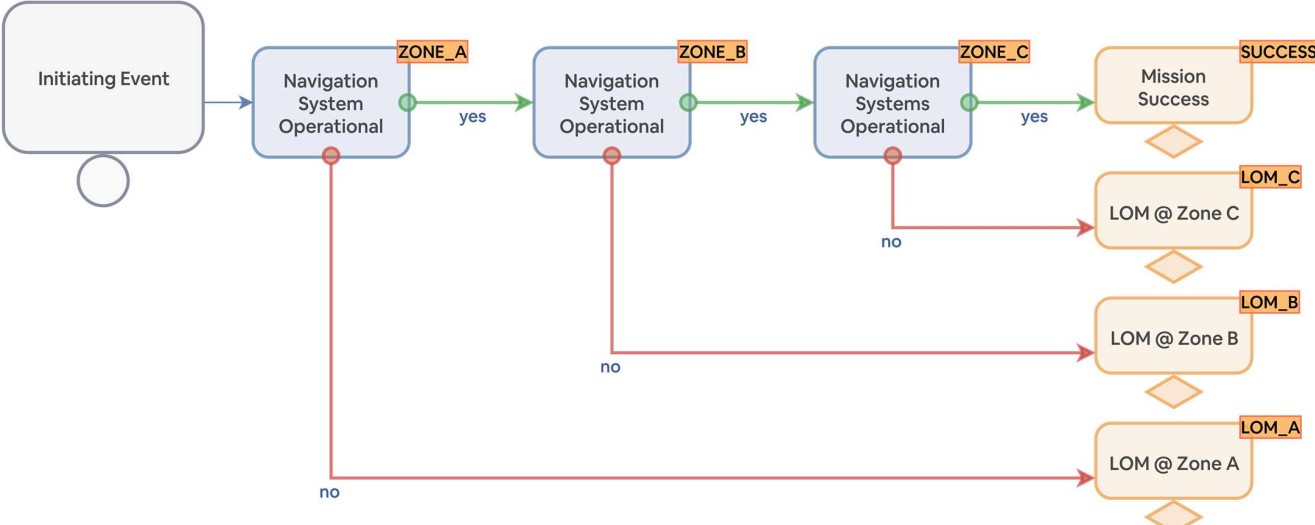

**Figure 4.** Event tree description of navigation system availability for radiation zones A, B, and C.

In addition, our model does not explicitly account for system recoveries or environmental factors beyond ambient radiation levels. While resilience is often associated with recovery capabilities after failures or disturbances, this aspect is not fully explored in this work. A DEPM-based analysis was presented in a previous work, which explicitly modeled the temperature dependence of digital systems on partial or full recovery per mission phase [29].

The model does not take into account the potential impact of weather conditions or terrain on the drone's ability to navigate each zone. These factors could significantly affect the drone's performance and likelihood of mission success. Lastly, our model does not factor in potential for human error in drone operation. In real-world scenarios, human error can significantly contribute to mission failure; however, incorporating such effects would add a layer of complexity that is beyond the scope of this study.

### 3.3. Navigation System Fault Tree

The fault tree for the drone's navigation system is constructed based on the potential failure modes of the system's components. This is depicted in Figure 5. A fault tree is a graphical representation of the logical relationships between the failures, or "basic events", and the system-level failure, or "top event". The basic events are the lowest level failures that can occur in the system, while the top event is the failure of the entire system. The intermediate events represent the failure of subsystems or groups of components. The logical relationships between these events are represented by gates, which can be "AND" gates, "OR" gates, or more complex logical gates.

In the given fault tree, the top event is the failure of the drone's navigation system, represented by the gate "TOP". This event can occur due to the failure of the power system "SYSTEM_POW", the positioning sensors "SENSOR_POS", the Kalman filter "FIL-TER_KAL", or the inertial measurement unit sensors "SENSOR_IMU". The intermediate events are represented by the gates "SENSOR_POS", "FILTER_KAL", "SYSTEM_POW", "SENSOR_GPS", "SENSOR_VIZ", "BATT_FAIL", and "GPS_SIGNAL". Each of these gates represents a failure mode that can contribute to the top event. For example, the "SEN-SOR_POS" gate represents the failure of the positioning sensors, which can occur due to the failure of the GPS hardware "GPS_HW" or the visual sensors "SENSOR_VIZ".

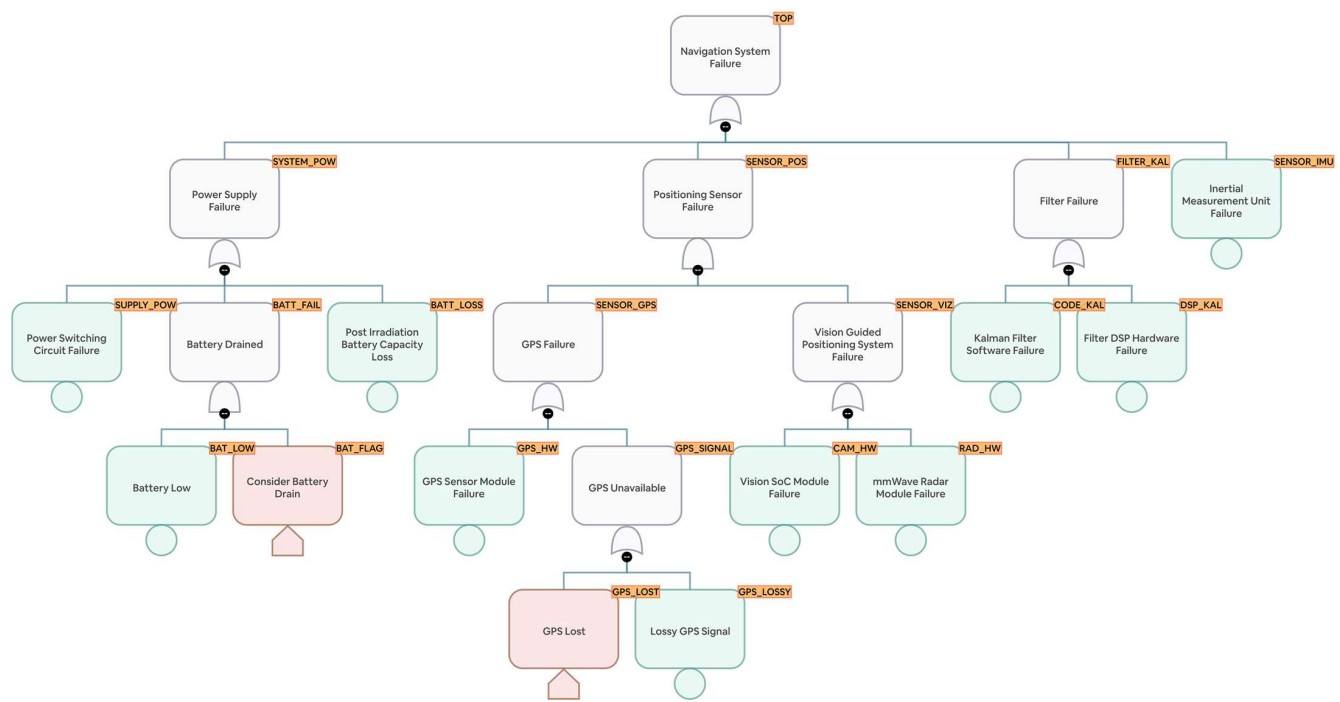

**Figure 5.** UAV navigation system failure fault tree.

The basic events are the lowest level failures that can occur in the system. These include the failure of the power supply "SUPPLY_POW", the battery running low "BAT_LOW", the loss of the battery "BATT_LOSS", the failure of the GPS hardware "GPS_HW", the loss of the GPS signal "GPS_LOSSY", the failure of the camera hardware "CAM_HW", the failure of the radiation sensor hardware "RAD_HW", the failure of the Kalman filter code "CODE_KAL", the failure of the Kalman filter DSP "DSP_KAL", and the failure of the IMU sensors "SENSOR_IMU". Failure rates for each hardware component and basic event are listed in Tables 6 and 7. These rates were acquired from the Texas Instruments reliability database [30]. They are used to calculate the probability of each basic event for the elapsed time at each radiation zone, which is used to calculate the probability of the intermediate and top events using the logical relationships defined by the gates. This allows for a quantitative assessment of the reliability of the drone's navigation system. Basic event BATT_LOW models battery drain, with the cumulative distribution function (CDF) plotted in Figure 6. We can observe that the battery was chosen to last well beyond the mission time.

**Table 6.** Manufacturer (Texas Instruments)-provided failure rates for generic drone hardware components.

| Basic Event | Part Number | Component Type | Derated Failure Rate [*failures/h*] |
|---|---|---|---|
| SENSOR_IMU | TI-MSP430 Series | MEMS IMU | $\mathcal{N}(\mu = 2.90 \,,\, \sigma \approx 2.00) \times 10^{-9}$ |
| CAM_HW | TI-TDA4AL-Q1 | Vision SoC + DSP | $\mathcal{N}(\mu = 2.10 \,,\, \sigma \approx 5.00) \times 10^{-9}$ |
| RAD_HW | TI-IWR1642AQAGABL | mmWave Radar + DSP | $\mathcal{N}(\mu = 3.80 \,,\, \sigma \approx 5.00) \times 10^{-9}$ |
| DSP_KAL | TI-TMS320C6678 | Kalman Filter DSP | $\mathcal{N}(\mu = 5.90 \,,\, \sigma \approx 3.50) \times 10^{-9}$ |

**Table 7.** Basic event probabilities for drone navigation system failure fault tree.

| Basic Event | Basic Event Description | Failure Rate [*failures/h*] |
|---|---|---|
| SENSOR_IMU | Inertial Measurement Unit Failure | $\mathcal{N}(\mu = 2.90\,,\ \sigma \approx 2.00) \times 10^{-9}$ |
| CAM_HW | Vision System-on-Chip Module Failure | $\mathcal{N}(\mu = 2.10\,,\ \sigma \approx 5.00) \times 10^{-9}$ |
| RAD_HW | mmWave Radar Module Failure | $\mathcal{N}(\mu = 3.80\,,\ \sigma \approx 5.00) \times 10^{-9}$ |
| DSP_KAL | Filter DSP Hardware Failure | $\mathcal{N}(\mu = 5.90\,,\ \sigma \approx 3.50) \times 10^{-9}$ |
| CODE_KAL | Kalman Filter Software Failure | DEPM, see section on Page 11 |
| GPS_HW | GPS Sensor Module Failure | $\mathcal{N}(\mu = 2.00\,,\ \sigma \approx 1.00) \times 10^{-9}$ |
| GPS_LOSSY | Lossy GPS Signal | $\mathcal{N}(\mu = 1.00\,,\ \sigma \approx 0.01) \times 10^{-6}$ |
| SUPPLY_POW | Switching Power Supply Circuit Failure | $\mathcal{N}(\mu = 1.00\,,\ \sigma \approx 0.50) \times 10^{-6}$ |
| BATT_LOW | Battery Low | Time-dependent; see Figure 6 |
| BATT_LOSS | Post-Irradiation Battery Capacity Loss | P = 1 as TID approaches TID limit |

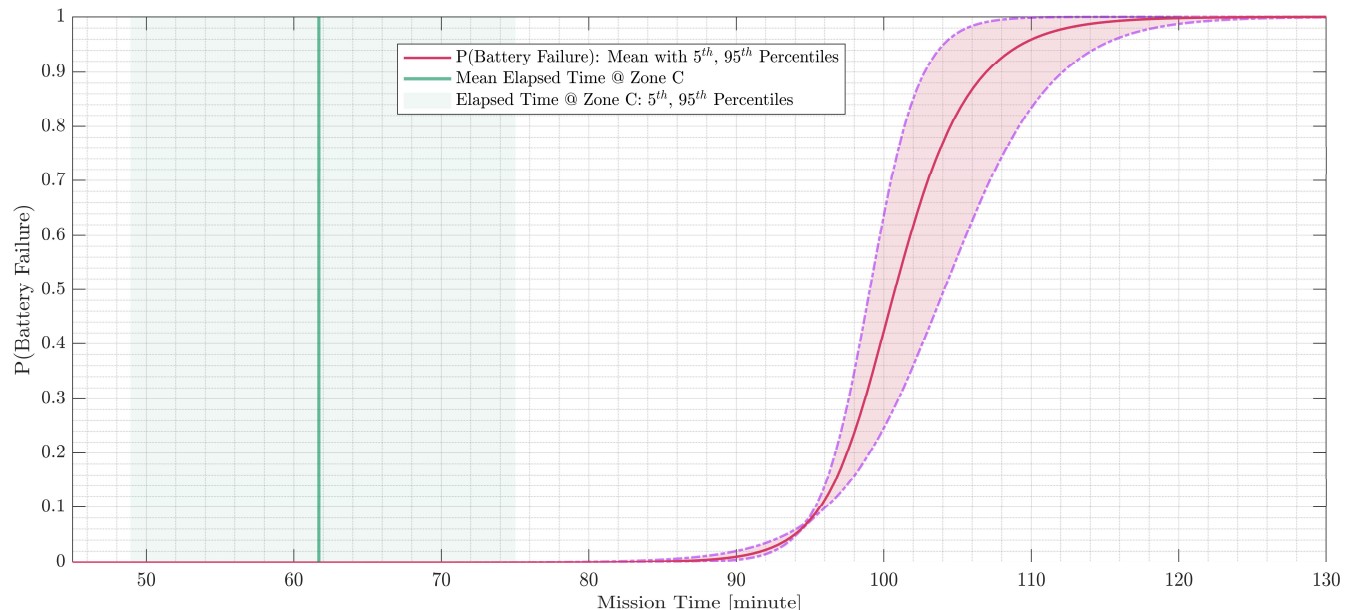

**Figure 6.** Cumulative distribution function for probability of battery drain for a range of mission times.

### 3.4. Modeling Kalman Filter Software Failures Using Dual-Graph Error Propagation Method (DEPM)

The Kalman filter software block is a critical component of the drone's navigation system. It processes the sensor data to estimate the drone's position and velocity, which are essential for controlling the drone's flight. The Kalman filter code is implemented on a digital signal processor (DSP), which is susceptible to radiation-induced single event upsets (SEUs) as well as TID effects. These events can cause bit flips in the processor's memory, leading to errors in the Kalman filter's calculations. In our model, we consider TID-related permanent failures. A detailed analysis of transient and accumulated failures in DEPM can be found in [29].

Failure analysis is performed by building a DEPM model from the Kalman filter assembly block, listed in Table 8. The Kalman filter algorithm is a recursive algorithm used to estimate the evolving state of a system. It consists of two main steps: the prediction step and the update step. The prediction step predicts the next state of the system and the update step corrects the prediction based on the actual measurement.

**Table 8.** Assembler code for single-variable Kalman filter algorithm.

```
/**
* Kalman Filter (single variable)
* Assume the input is in register R0
* Assume the initial state estimate is in register R1
* Assume the initial error covariance is in register R2
* Assume the process noise variance is in register R3
* Assume the measurement noise variance is in register R4
**/
```

**Initialization**

| | | |
|---|---|---|
| 1. | MOV R5, R1 | // R5 = State estimate (copy of initial state estimate) |
| 2. | MOV R6, R2 | // R6 = Error covariance (copy of initial error covariance) |
| 3. | LOOP: | |

**Prediction step**

| | | |
|---|---|---|
| 4. | MOV R7, R5 | // R7 = Predicted state estimate (copy of state estimate) |
| 5. | ADD R5, R7 | // R5 = State estimate = state estimate + predicted state estimate |
| 6. | MOV R8, R6 | // R8 = Predicted error covariance (copy of error covariance) |
| 7. | ADD R8, R3 | // R8 = Predicted error covariance + process noise variance |
| 8. | MOV R6, R8 | // R6 = Error covariance = predicted error covariance + process noise variance |

**Update step**

| | | |
|---|---|---|
| 9. | MOV R9, R6 | // R9 = Error covariance (copy of error covariance) |
| 10. | ADD R9, R4 | // R9 = Error covariance + measurement noise variance |
| 11. | MOV R10, R9 | // R10 = Temporary variable for division |
| 12. | DIV R8, R10 | // R8 = Kalman gain = error covariance/(error covariance + measurement noise variance) |
| 13. | MOV R11, R0 | // R11 = Measurement |
| 14. | SUB R11, R7 | // R11 = Measurement − predicted state estimate |
| 15. | MUL R11, R8 | // R11 = Innovation = (measurement − predicted state estimate) × Kalman gain |
| 16. | ADD R5, R11 | // R5 = State estimate = state estimate + innovation |
| 17. | MOV R12, R8 | // R12 = Kalman gain (copy of Kalman gain) |
| 18. | SUB R12, R8 | // R12 = 1 − Kalman gain |
| 19. | MUL R6, R12 | // R6 = Error covariance = error covariance × (1 − Kalman gain) |

*// Continue the loop or terminate*

In the DEPM, the assembly code is first translated into a control flow graph (CFG) and a data flow graph (DFG). The CFG represents the flow of control in the program, while the DFG represents the flow of data between operations. The DEPM then combines these two graphs into a dual graph, which represents both the control flow and data flow in the program. The DEPM is used to analyze the propagation of accumulated errors in the software, caused by TID effects in the DSP hardware. Figure 7 illustrates the DEPM for the Kalman filter assembly in Table 8, compiled using the LLVMPars framework [21,31].

*3.5. Modeling Total Ionizing Dose Limits for Electronic Hardware*

TID is a measure of the amount of radiation absorbed by electronic components. Excessive TID can cause degradation or failure of these components, leading to mission failure. In order to assess the availability of the drone's electronic hardware in each radiation zone, the TID limits for each component need to be determined. The TID limits for electronic components are typically provided by manufacturers and are based on the radiation hardness of the components [32]. These limits specify the maximum TID that a component can withstand without experiencing significant degradation or failure. TID limits for these components are consolidated from the manufacturer's specifications or empirical tests and are listed in Table 9, sources for which can be obtained by the following corresponding references.

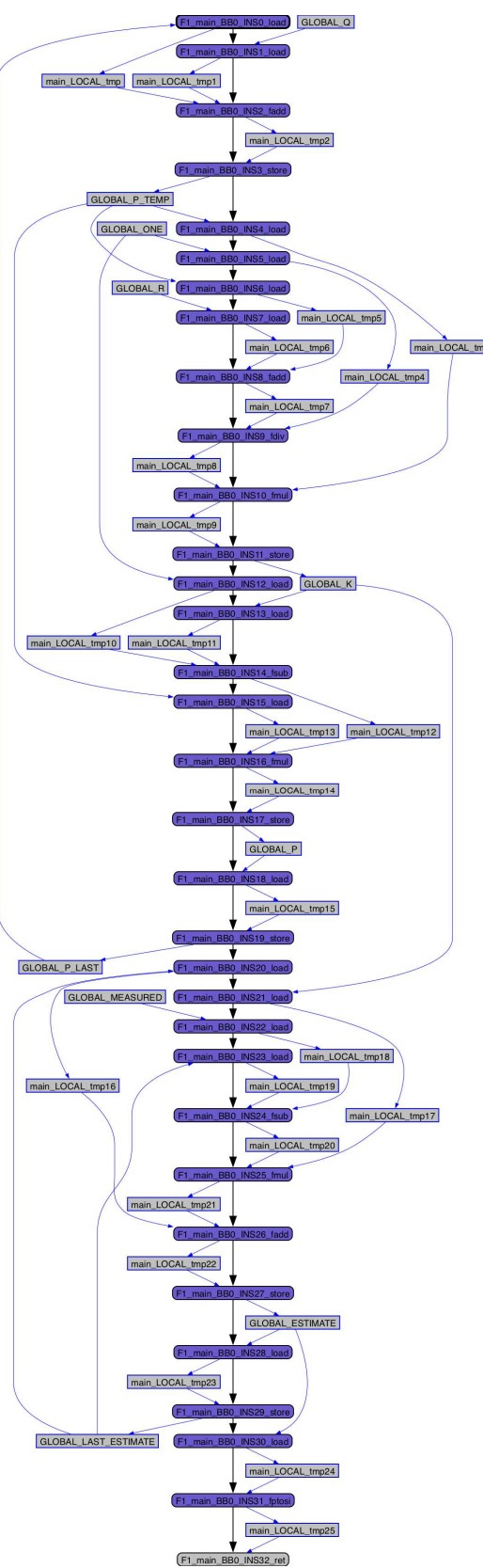

**Figure 7.** Dual-graph error propagation method (DEPM) representation of assembly for a single-variable Kalman filter.

**Table 9.** Total ionizing dose (TID) limits, COTS components.

| Component | Commercial Off-The-Shelf (COTS) |
|---|---|
| Inertial Measurement Unit | $\mathcal{U}(min = 1.00\,,\ max = 5.50) \times 10^4$ |
| Power Switching Circuit | $\mathcal{U}(min = 1.50\,,\ max = 2.00) \times 10^4$ |
| Lithium Ion Battery | $\mathcal{U}(min = 0.10\,,\ max = 2.74) \times 10^6$ |
| GPS Sensor Module | $\mathcal{U}(min = 1.43\,,\ max = 1.74) \times 10^4$ |
| Vision SoC Module | $\mathcal{U}(min = 0.10\,,\ max = 1.00) \times 10^4$ |
| mmWave Radar Module | $\mathcal{U}(min = 0.10\,,\ max = 1.00) \times 10^4$ |
| Filter DSP Hardware | $\mathcal{U}(min = 0.10\,,\ max = 1.00) \times 10^4$ |

Using the TID limits from Table 9, the probability of failure for each component can be calculated based on the total received dose. This probability is then used to determine the likelihood of component failure in each radiation zone. For example, consider the COTS Inertial Measurement Unit (IMU) component. The TID limit for the IMU is in the range of $\mathcal{U}(min = 1.00\,,\ max = 5.50) \times 10^4$ [33]. Based on the total received dose in each radiation zone, the probability of exceeding the TID limit for the IMU can be calculated. If the probability of exceeding the TID limit is below this threshold, the IMU is considered to have survived in that radiation zone. Otherwise, the IMU is considered to have failed. Similarly, the probability of failure can be calculated for other components, such as the power-switching circuit, lithium-ion battery [34], GPS sensor module, vision SoC module, mmWave radar module, and filter DSP hardware [35,36].

Components can be targeted for radiation-hardening measures, such as shielding or the use of radiation-hardened components, to improve their availability in nuclear-contaminated environments. In the next section, we present the results of this analysis, and re-run it after radiation-hardening the chosen components.

## 4. Results and Discussion

The results of the event sequence analysis for the drone's navigation system are presented in this section. The modeling, quantification and visualization were performed using the OpenPRA framework, which integrates the event tree/fault tree approach with DEPM. OpenPRA is under active development and this analysis represents its current modeling capability. The results are presented in terms of the LOM likelihood in each radiation zone.

### 4.1. Probability of Loss of Mission (LOM) Using Commercial Off-The-Shelf (COTS) Components

The probability of LOM in each radiation zone is calculated based on the failure probabilities of the components in the drone's navigation system. The results are plotted in Figure 8 and listed in Table 10. As expected, the probability of LOM increases with the radiation level, with the highest probability occurring in Zone C, the highest radiation zone. This is due to the higher TID received by the components in this zone, which increases the likelihood of component failure. There are no TID-related failures in Zones A and B. To highlight the contribution from TID failures, the Zone C distribution was split into two.

**Table 10.** Probability of loss of mission (LOM) due to COTS drone navigation system failure.

| End State | End State Description | P (End State) |
|---|---|---|
| LOM_A | Loss of Mission in Zone A | $\mathcal{LN}(\overline{m} \approx 1.35 \times 10^{-7},\ EF \approx 6.29)$ |
| LOM_B | Loss of Mission in Zone B | $\mathcal{LN}(\overline{m} \approx 1.20 \times 10^{-6},\ EF \approx 3.56)$ |
| LOM_C | Loss of Mission in Zone C | $\mathcal{LN}(\overline{m} \approx 1.39,\ EF \approx 0.35) \times 10^5$ |
| SUCCESS | Mission Success | $\mathcal{LN}(\overline{m} \approx 2.70 \times 10^{-1},\ EF \approx 10.5)$ |

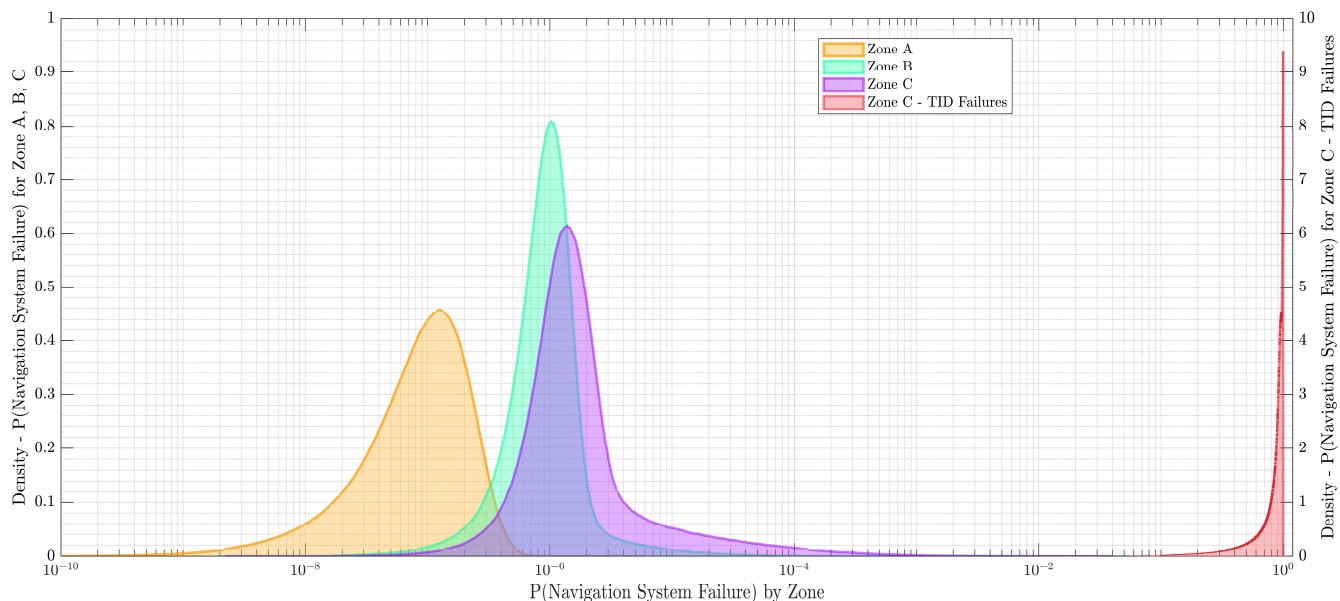

**Figure 8.** Density estimates of COTS navigation system failure probabilities by zone and TID effects.

Table 10 presents the probability of LOM due to the failure of the drone's navigation system in each radiation zone (A, B, and C). The sampled probabilities are parametrized using a log-normal distribution (LN), with the mean (m) and error factor (EF) parameters provided. The probability of LOM is low and dependent on non-radiation-related phenomena for all parts of Zones A and B, and most parts of Zone C. This suggests that the drone's navigation system is relatively reliable in low-radiation environments, averaging about one LOM per ten thousand missions. However, the probability of LOM increases significantly in Zone C. This is due to the higher TID received by the components in this zone, which increases the likelihood of component failure. This is a significant concern, as it suggests that the drone may not be able to complete its mission in high-radiation environments. This could have serious consequences for SAR missions, as it could prevent the drone from reaching survivors or accurately assessing the extent of the damage.

In terms of mission success, the results indicate a relatively low probability. This suggests that the current design of the drone's navigation system may not be suitable for SAR missions in nuclear-contaminated environments. Therefore, improvements to the design, such as the use of radiation-hardened components or shielding, may be necessary to increase the probability of mission success.

### 4.2. Selective Radiation-Hardening Using Mission Success Criteria

With the objective of improving the unacceptably low likelihood of mission success, we propose a strategy to selectively harden components from the navigation system. We begin by choosing a component and assign a wide distribution for its TID limit. For instance, we choose the DSP and assign its TID limit as $TID_{DSP} = \mathcal{LU}(min = 1.0 \times 10^0,\ max = 1.00 \times 10^6)$. Here, $TID_{DSP}$ is a loguniform distribution, and much wider than the nominal value specified in Table 9. Next, we invert the probability of mission success, making it conditional on the event $TID_{DSP}$, and accept $TID_{DSP}$ values only when LOM does not occur. Figure 9 plots the kernel density estimates for the sampled, accepted and rejected DSP TID limit ranges at the 95th percentile for 1 in 10,000 mission failures. By extension, sampling over a range of expected mission failure rates, we can construct a radiation-hardening vs. mission failure curve. This curve is illustrated in Figure 10.

$$P_S(TID) = P(TID_{DSP} \mid \neg LOM) \tag{4}$$

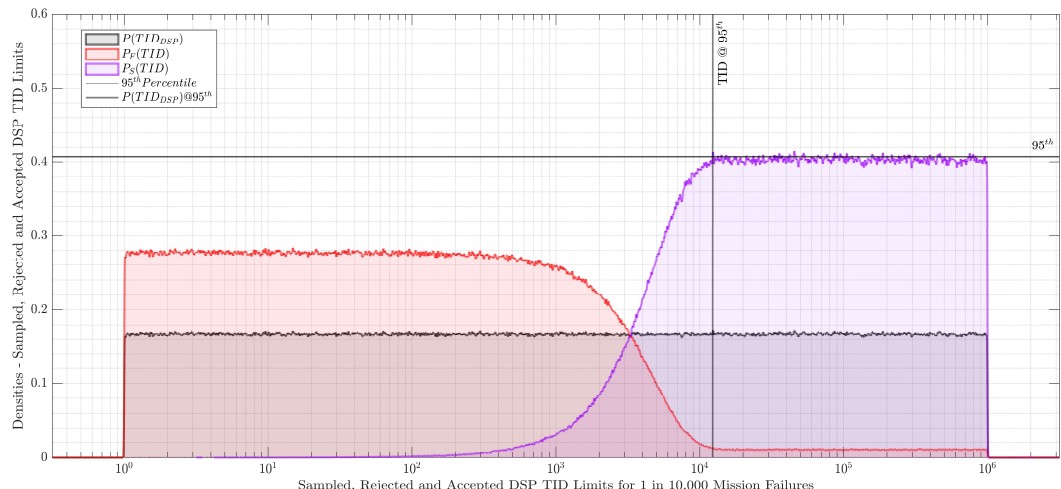

**Figure 9.** Digital signal processor total ionizing dose limits for 1 in 10,000 mission failures.

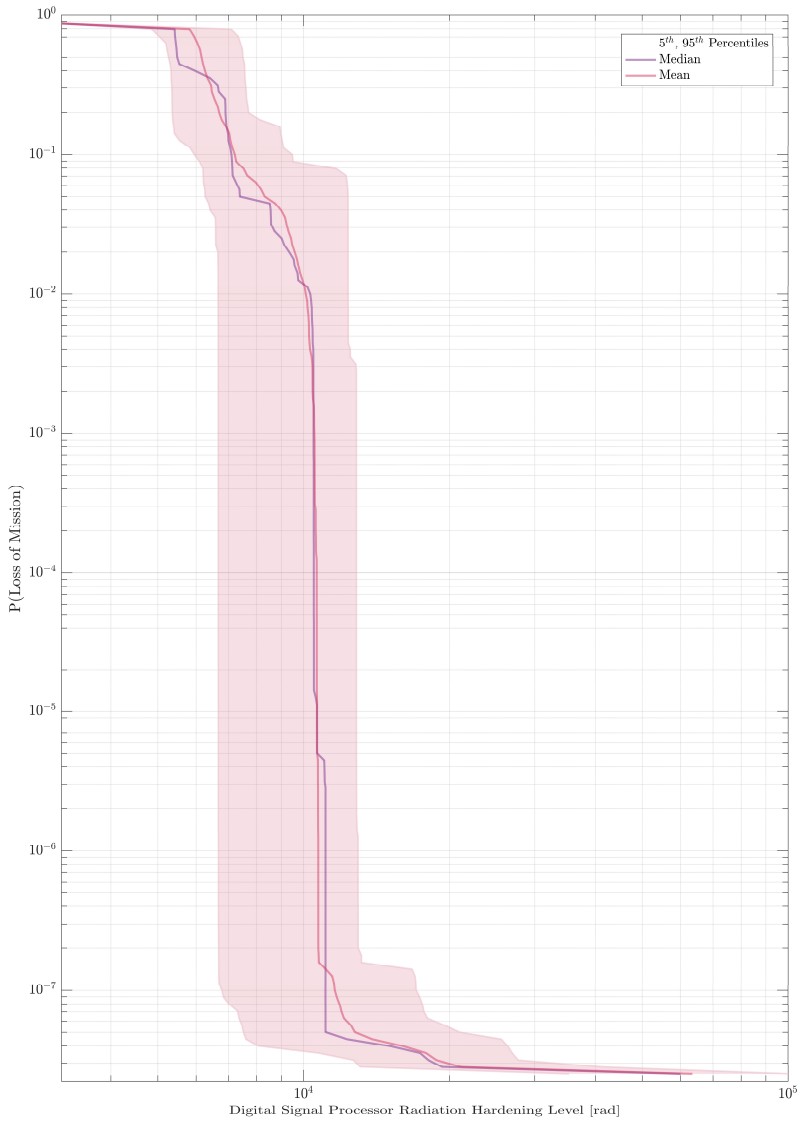

**Figure 10.** Radiation-hardening targets vs. likelihood of mission failure.

The results of the analysis allow for us to choose a radiation-hardening limit based on target mission success criteria.

## 5. Conclusions

This study presented a probabilistic risk assessment (PRA) approach for assessing the reliability of commercial off-the-shelf (COTS) drones in nuclear-contaminated search and rescue (SAR) missions. The approach integrated the event tree/fault tree digraph method with the dual-error propagation method (DEPM) to model potential loss of mission (LOM) scenarios due to combined hardware-software failures in the drone's navigation system. The impact of radiation on the drone's components was simulated by comparing the total ionizing dose (TID) with the acceptable limit for each component. To mitigate TID-based component failure, a strategy was presented for selectively hardening components based on the desired mission success criteria.

The results of this analysis can aid in the integration of COTS components into radiation-hardened designs, optimizing the balance between cost, performance, and reliability in drone systems for nuclear-contaminated SAR missions.

While the study provides valuable insights into the reliability of COTS drones in nuclear-contaminated environments, it also has several limitations. The analysis only considered the availability of the navigation system and did not take into account all components of the UAV and their interdependencies, including common-cause failures. The model also did not consider the potential impact of weather conditions or terrain on the drone's ability to navigate each zone, or the potential for human error in drone operation. Furthermore, the study only considered TID effects and did not take into account other failure mechanisms such as displacement damage and single-event effects (SEEs). In addition, the mission objective was limited to the failure of a single drone, which is not fully representative of fleet-based systems.

Future work will address these limitations by developing a more comprehensive model that includes all components of the UAV and their interdependencies and considers other failure mechanisms and environmental factors. Additionally, future research will explore the use of dynamic PRA methods and integrate them into a framework that collectively assesses the cost, performance, and reliability of drone systems for nuclear-contaminated SAR missions. Additionally, we will explore the broader context of resilience, including recoveries, evolving mission objectives, and the occurrence of unspecified failures. This will provide a more holistic understanding of the resilience of drone systems in nuclear-contaminated environments.

**Author Contributions:** Conceptualization, A.E. and M.A.D.; methodology, A.E. and M.A.D.; verification, A.E. and M.A.D.; formal analysis, A.E.; investigation, A.E. and M.A.D.; resources, A.E. and M.A.D.; data curation, A.E.; writing—original draft preparation, A.E.; writing—review and editing, M.A.D.; visualization, A.E.; supervision, M.A.D.; project administration, M.A.D.; funding acquisition, M.A.D. All authors have read and agreed to the published version of the manuscript.

**Funding:** This research received no external funding.

**Data Availability Statement:** Simulation methods and data for this study are available at https://gitlab.openpra.org/published-tools-and-methods/integrating-commercial-off-the-shelf-components-into-radiation-hardened-drone-designs-for-nuclear-contaminated-search-and-rescue-missions (accessed on 1 August 2023).

**Conflicts of Interest:** The authors declare no conflict of interest.

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
