# Peer review of "Integrating Commercial-Off-The-Shelf Components into Radiation-Hardened Drone Designs for Nuclear-Contaminated Search and Rescue Missions"

_drones, doi:10.3390/drones7080528_

Round 1

Reviewer 1 Report

The paper generally leaves a positive impression, since it discusses the topical issue of failures of commercial drones (although related to their usage in rather rare circumstances). The quality of presentations of results is good, altough I'd recommend replacing the images to reduce the blurring effects (consider using vector formats), and increasing the font sizes for the text in the graphs.

My main concern related to the manuscript is that, in my opinion, the scientific novelty and originality of the contribution is not highlighted clearly enough. The authors claim a new approach called D-PRA which is based on some earlier known tecniques developed by other researchers, and the structure of the paper makes it hard to understand where the already developed tools end and the authors' contribution begins. Particularly:

- The starting point is the authors' statement that "additional dynamic methods within PRA models can lead to improved accuracy", which implies they add one into PRA. In the abstract it is said that "The D-PRA is based on a discrete-dynamic event tree (D-DET) approach, which couples with the OpenEPL error propagation framework". At the same time, they do not mention neither D-DET, nor discrete dynamic trees, anywhere in the following text. If the synonym is used (e.g., "event tree/fault tree approach"), then it should be explained.

- Line 297: "The D-PRA was performed using the OpenPRA framework". This framework is not the authors' result, does it mean that the whole D-PRA technique was also implemented earlier, so it cannot be named the scientific novelty?

- All in all, the described approach seems a particular example of application of Markov processes. Indeed, in line 119 the authors mention that "we transform our DEPM models into discrete-time Markov Chains (DTMCs) or continuous time Markov Chains (CTMCs)". Thus the question arises why one needs to distinguish DEPM and DTMC/CTMC. Or is there something else in D-PRA approach which does not fit into MC?

- Last but not least, there is no review with the examples of using similar techniques by other research teams and their drawbacks, thus the necessity of developing new approaches (contrary to application of the known approaches to an important problem, such as drone failure) is not entirely clear.

These issues result in the fact that the "Conclusion" section seems somewhat unconvincing. The domain-specific results claimed in it are rather trivial (e.g., "probability of loss of mission (LOM) increases with the radiation level"). The obtained quantitative result, which includes the values for LOM probability, is more promising but it is scarcely analyzed for correctness (I would suggest uncertainty and sensitivity analsysis). And the novelty of the framework itself, as I mentioned before, is not well-justified.

I recommend a major revision of the paper so that the edited text will be able to address the abovementioned concerns.

Sometimes the authors tend to use overcomplicated sentences, e.g. including a handful of nouns (to name a few, "blocks to quantify navigation system availability per radiation zone" and "to the extent specified by a predefined mission success criterion"). Such sentences make it harder to grasp the meaning and should be avoided.

Reviewer 2 Report

The purpose of this study is to address the challenges and risks associated with using commercial-off-the-shelf (COTS) drones in nuclear-contaminated search and rescue (SAR) missions. The specific purposes of this study include:

·       Assessing Survivability: The study aims to evaluate the survivability of COTS drones in radiological SAR missions. It involves analyzing the impact of high radiation levels on the electronic components of the drones, which can potentially lead to mission failure .

·       Dynamic Probabilistic Risk Assessment (D-PRA): The study presents a dynamic probabilistic risk assessment approach to evaluate the survivability of COTS drones. This approach involves modeling time-dependent risks using Kripke structure notation and resilience ontology terms .

·       Identifying Vulnerable Components: The study aims to identify the most vulnerable components of COTS drones in radiation zones. By identifying these components, the study can provide insights into potential improvements for radiation hardening (RAD-HARD) to enhance the reliability and safety of the drones ,

·       Enhancing Effectiveness in SAR Missions: By improving the reliability and safety of COTS drones in radiation zones, the study aims to enhance their effectiveness in nuclear-contaminated SAR missions. This can contribute to better assessment of damage, locating survivors, and monitoring radiation levels in the aftermath of a nuclear accident , .

Overall, the purpose of this study is to provide a systematic and comprehensive framework for assessing the survivability of COTS drones in nuclear-contaminated SAR missions and to enhance their reliability, safety, and effectiveness in such environments .

Comments:

1.     There are no previous works. Please add section after introduction section call Related works and discuss the close previous works to the proposed work.

2.     The authors should make comparison table in the end of related works. The table should clarify the difference between the proposed work and previous works.

3.     The authors should revise figure five "Figure 5. Navigation system failure fault tree". Unclear resolution.

4.     3.    The authors should revise figure six “Figure 6. Cumulative Distribution Function for Probability of Battery Drain for a range of mission 235 times”. Unclear resolution.

5.     The authors should revise figure seven "Figure 7. Dual Graph Error Propagation (DEPM) representation of assembly for a single variable 294 Kalman filter." by make it partitions to be clear and readable.

6.     The authors should revise figure ten "Figure 10. Radiation Hardening Targets vs Likelihood of Mission Failure". Unclear resolution.

7.     The authors should make evaluations with close previous works.

8.     Survivability Assessment:

·       The specific system properties, events, and emergent system behaviors that are being defined and analyzed in the survivability assessment are not clearly stated .

·       The study does not provide information on the methodology or approach used for the survivability assessment, leaving it unclear how the assessment was conducted.

9.     Dynamic Probabilistic Risk Assessment (D-PRA):

·       The study does not provide a detailed explanation of the Dual Graph Error Propagation (DEPM) representation and how it is applied in the assembly for a single variable Kalman filter .

·       The study does not elaborate on the results and discussion related to the D-PRA approach, leaving it unclear what specific findings or conclusions were drawn .

10.  Identification of Vulnerable Components:

·       The study does not provide information on the criteria or methodology used to identify vulnerable components for radiation hardening improvements.

·       It is unclear how the study determines the criticality of components and their susceptibility to radiation effects.

11.  Enhancing Effectiveness in SAR Missions:

·       The study does not provide specific details on the methods or strategies proposed to enhance the effectiveness of COTS drones in nuclear-contaminated SAR missions.

·       The study does not elaborate on the results and discussion related to enhancing effectiveness, leaving it unclear what specific findings or recommendations were made.

12.  Methodology and Approach:

·       Provide a detailed explanation of the methodology and approach used for the survivability assessment, including the specific system properties, events, and emergent system behaviors being defined and analyzed.

·       Elaborate on the Dual Graph Error Propagation (DEPM) representation and its application in the assembly for a single variable Kalman filter, providing a clear understanding of how errors are propagated in the drone's navigation system.

13.  Results and Discussion:

·       Expand on the results and discussion related to the Dynamic Probabilistic Risk Assessment (D-PRA) approach, providing specific findings and conclusions drawn from the assessment.

·       Include a comprehensive analysis of the vulnerabilities and weaknesses identified in the drone's navigation system, highlighting the most critical components for radiation hardening improvements.

14.  Vulnerable Component Identification:

·       Describe the criteria or methodology used to identify vulnerable components for radiation hardening improvements, providing a clear understanding of how criticality and susceptibility to radiation effects are determined.

·       Consider incorporating additional factors, such as single-event effects (SEEs) and displacement damage, into the vulnerability assessment to enhance its comprehensiveness.

15.  Enhancing Effectiveness in SAR Missions:

·       Provide specific details on the proposed methods or strategies to enhance the effectiveness of commercial off-the-shelf (COTS) drones in nuclear-contaminated SAR missions.

·       Consider addressing the impact of weather conditions and terrain on the drone's ability to navigate each radiation zone, as well as incorporating the potential for human error in drone operation into the model.

16.  Limitations and Future Work:

·       Clearly outline the limitations of the study, including any assumptions made or potential sources of uncertainty.

·       Identify areas for future research and development, such as extending the D-PRA approach to include other potential failure modes, considering the impact of weather conditions and terrain, and incorporating human error in drone operation into the model.

17.  There are some English mistakes that can be identified in the study:

·       In line 371, "comprehen- sive" should be "comprehensive."

·       In line 373, "ap- proach" should be "approach."

·       In line 375, "their" should be "its" to maintain consistency with "drones" as a singular noun.

18.  The authors should make table for evaluation parameters used in the simulation.

19.  What are the limitations of proposed method? Please answer the question in the article.

20.  Please  follow the journal template.

Reviewer 3 Report

The manuscript studied the Dynamic Probabilistic Risk Assessment of COTS drones. In my opinion, the authors did not present the methodology clearly, it is not easy to realize the method for authors. Additionally, which part of the method reflects the characteristic “dynamic”? I also think verification is lacking in the manuscript, the authors should compare the proposed method with the present research.

The english should be improved.

Reviewer 4 Report

1) Authors write that they propose Dynamic Probabilistic Risk Assessment (D-PRA) approach. It should be noted that such an approach and corresponding methods were developed quite a long time ago, therefore it is necessary to expand the analysis and clearly define the features or the use of a known approach, or the differences (methodological and mathematical) of own approach.

A couple of appropriate links:

https://link.springer.com/chapter/10.1007/978-3-030-88098-9_2

https://www.researchgate.net/publication/358596065_Dynamic_Probabilistic_Risk_Assessment_of_Complex_Systems

2) Concept of systems should be clarified. It means a set of components for single drone? Usually for such mission in case of severe accidents on the NPPs or other critical objects a fleet (or swarm) of drones can/should be applied. Hence, system can consists of a set of UAVs. There are a lot of publications for such systems describing models of UAV fleets reliability considering different routes, conditions of applications and so on.

Hence reliability model of single UAV is a partial case of UAV fleet model… Please clarify this issue.

3)      Methodology (section 2) should be presented more clear and explanation how different notations and models are composed to get the final result. Is not enough clear how Markov chain (section 2, figure 2) is applied?

4)      Is not understandable how reliability of software based components is taken into account considering other nature and reliability models of such components.

5)      It’s needed to explain how Figure 7 (integrated algorithm) has been developed/composed in detail.

6)      Usually if COTS components are applied in critical systems (for space, nuclear and so on) grounding of such approach is added by general assessment of efficiency according with criterion reliability –cost (acceptable risk or required probability of up-state and minimal cost). What about this criterion part of research? 

7) Minor: authors use a few attributes for UAV: reliability, resilience, survivability... Please specify their using.

No comments

Round 2

Reviewer 1 Report

I am satisfied with the modifications made according to my recommendations, as well as the comments of the authors, so I can recommend the paper for publication.

Author Response

Thank you for your support.

Reviewer 3 Report

I do not think the modificaiton is enough, the novelty is still lacking in the manuscript.

I do not think the modificaiton is enough, the novelty is still lacking in the manuscript.

Author Response

Thank you for your review.

Reviewer 4 Report

The paper has been revised considering the remarks and recommendations. 

Authors interpret the concept of resilience somewhat simplistically, since its scope is far from being exhausted by the possibility of full or partial recovery of the system after failures. This vision could be further explained taking into account the wider context (the possibility of adaptation in case of physical and cyber impacts, changing requirements, occurrence of unspecified failures...).
